# Density-Scaled Regularization for Offline Reinforcement Learning

## Abstract

Value-based offline RL methods are prone to overestimate the values of out-of-distribution (OOD) actions, and this is often addressed by regularizing the action-value function in the Bellman update. However, existing regularization methods can suffer from being too conservative, which can arise from over-penalizing the values for both in-distribution actions and out-of-support actions. We present a new regularization method for offline value-based methods, called Density-Scaled (DS) regularization, which penalizes the value function based on the relative action density of the behavior policy. We show a theoretical connection between our method and the existing Supported Value Regularization (SVR) method, demonstrating how the SVR solution for policy evaluation can be viewed as a limiting case of the solution from the DS regularized problem. Empirical results demonstrate that the DS penalty is competitive with the state-of-the-art techniques, and allows greater flexibility in the estimation of the behavior policy compared to SVR due to improved robustness and numerical stability.

## 1 Introduction

Offline reinforcement learning (offline RL) studies how an agent can learn an optimal policy for sequential decision-making from a dataset of a (typically suboptimal) policy's interactions with the environment (Levine et al., 2020). While standard online RL requires the agent to iteratively collects data through trial-and-error interaction and use this experience to improve its behavior over time, direct interaction with the environment can be expensive, unsafe, or impractical, for many real-world applications, such as healthcare, robotics, or recommendation systems. Offline RL addresses this limitation by learning policies solely from a fixed dataset of previously collected transitions, without further environment interaction. By leveraging pre-existing logs of experience, offline RL enables data reuse and avoids the risks associated with exploratory behavior in safety-critical domains.

Nevertheless, learning from static datasets introduces unique challenges, particularly due to the distributional mismatch between the learned policy and the behavior policy that generated the data (Fujimoto et al., 2018; Levine et al., 2020). Typical model-free RL algorithms estimate a value function using Bellman backups and subsequently derive a policy by maximizing these estimated values. However, in the offline setting, the learned policy may query the value function on actions that are poorly represented or entirely absent in the dataset. Function approximation can therefore extrapolate arbitrarily to these out-of-distribution (OOD) actions, and the maximization in the Bellman operator may amplify such errors, resulting in systematically over-optimistic value estimates that propagate through bootstrapping (Kumar et al., 2019; Fujimoto et al., 2019).

To improve generalization beyond experiences in the offline dataset, prior works have proposed several forms of regularization (Kostrikov et al., 2021a; Kumar et al., 2020; Wu et al., 2019; Xu et al., 2023), as discussed in more detail in Section 2. We focus on value-regularization approaches, which attempt to directly control extrapolation by enforcing conservative or pessimistic value estimates for OOD actions. A representative approach is Conservative Q-Learning (CQL) (Kumar et al., 2020), which adds an explicit regularization term that broadly penalizes the value of all actions proposed by the learned policy. This can lead to overly

pessimistic value estimates since actions within the data distribution are also penalized, and the specific form of the penalty can in theory lead to unbounded values. On the other hand, support-based methods, such as the state-of-the-art Supported Value Regularization (SVR) (Mao et al., 2023), aim to penalize the value of actions outside the support of the behavior policy without affecting the in-support actions. While this is empirically more effective than CQL and prior methods, SVR does not consider the relative uncertainty of value estimates from the action density in the dataset – for example, sparsely supported regions of the action space are not penalized any more than well-supported regions. These contrasting approaches highlight the trade-off between pessimism and generalization: pessimistic penalties that consider all actions can risk over-conservatism, whereas strict support constraints can forgo penalizing estimates in poorly-supported regions that may benefit from conservatism.

In this work, we provide a different approach for managing this tradeoff that combines advantageous aspects of both paradigms. By comparing the update solutions for CQL and SVR, we gain insight into their theoretical advantages and limitations, and devise a new regularization method that is more nuanced with respect to in-distribution and out-of-support actions. Our approach, termed Density-Scaled (DS) regularization, is a penalty that varies the level of conservatism with the density of actions in the dataset. While our penalty term may appear simple and heuristic at first, we analyze the solution and establish an interesting interpolating property that connects it to the existing support-constraint solution of SVR, and results in improved numerical stability compared to it. In summary, we make the following contributions:

- We introduce the Density-Scaled regularizer for offline policy evaluation, which penalizes the value function smoothly based on an estimated model for the behavior density. It can be implemented and applied in a straightforward manner for many model-free offline RL methods.

- We provide theoretical analysis for our method, proving that our penalty produces conservative estimates that lower bound the true Q-function. We also demonstrate that the solution to our regularized Bellman update resembles a generalized version of the SVR solution.

- We empirically demonstrate the effectiveness of our method across offline RL benchmarks and visual datasets with high-dimensional pixel observations. Furthermore, our ablation studies demonstrate that this penalty is more robust to inaccurate estimation of behavior policy compared to SVR.

## 2 Related Work

Our work falls under model-free offline RL with value-based learning. In this setting, value regularization and policy regularization are two commonly used approaches for alleviating the extrapolation error.

Value regularization methods aim to alleviate overestimation errors in critic learning. In-sample or in-support learning methods develop value learning objectives that avoid querying unseen actions in the Bellman backup altogether (Kostrikov et al., 2021a; Mao et al., 2024; Xu et al., 2023). Ensemble-based methods reduce overestimation by taking the minimum over multiple $Q$-functions (Agarwal et al., 2020; Kumar et al., 2019), or by incorporating uncertainty quantification into the target (O'Donoghue et al., 2018; Wu et al., 2021; An et al., 2021). A major class of approaches directly penalizes Q-values during training, typically by adding a conservative regularization term (Kumar et al., 2020; Kostrikov et al., 2021b; Mao et al., 2023; Shimizu et al., 2024). these methods do not explicitly identify the OOD region but instead induce conservatism based on current policy and behavior policy density. In contrast, support-based regularization (Mao et al., 2023; Wu et al., 2022) aim to explicitly constrain the values of the value function on out-of-support regions. Our method falls under the class of explicit regularization methods. The recently proposed Flow Actor-Critic (Chae et al., 2026) is most similar to our work, where the behavior density is explicitly estimated with a flow model and the Q-function is penalized in proportion to a thresholded weighting term derived from it. Our method differs by considering a quadratic penalty that results in lower-bounded Q-functions, and also explicitly considers the normalized density which ensures that the penalty is theoretically bounded without additional thresholding.

Complementary to value regularization, policy regularization methods constrain the learned policy to remain close to the behavior policy, which prevents the Q-function from being evaluated on out-of-distribution

actions, reducing errors in the Bellman update. Policy constraints are typically implemented via divergence penalties, such as KL or MMD regularization (Wu et al., 2019; Kumar et al., 2019; Jaques et al., 2019), or behavior cloning terms, such as TD3-BC (Fujimoto & Gu, 2021) and ReBRAC (Tarasov et al., 2023). Another line of work known as one-step RL considers learning a policy with single regularized update step (Brandfonbrener et al., 2021; Park et al., 2025; Peng et al., 2019) Our method is compatible with existing policy regularization approaches and we evaluate instances where both policy regularization and the DS penalty is used simultaneously.

## 3 Preliminaries

Markov Decision Processes (MDPs) provide a general mathematical framework for sequential decision-making. An MDP is defined by a tuple $M = (S, A, P, R, \gamma)$, where $S$ is the state space, $A$ the action space, $P(s' \mid s, a)$ the transition probability distribution for the dynamics of the system, $R(s, a)$ the reward function, and $\gamma \in [0, 1)$ the discount factor. The objective is to find the optimal policy $\pi^*$ that maximizes the expected cumulative reward $J(\pi) = \mathbb{E}_\pi \left[ \sum_{t=0}^\infty \gamma^t r(s_t, a_t) \mid s_0 \sim \mu_0 \right]$, where the expectation under $\pi$ means $a_t \sim \pi(\cdot \mid s_t), s_{t+1} \sim P(\cdot \mid s_t, a_t)$, and $\mu_0$ is the initial state distribution. Given a policy $\pi$, the state-action value function $Q^\pi(s, a)$ represents the expected cumulative reward when taking action $a$ in state $s$ and then following policy $\pi$: $Q^\pi(s, a) = \mathbb{E}_\pi \left[ \sum_{t=0}^\infty \gamma^t r(s_t, a_t) \mid s_0 = s, a_0 = a \right]$. The Q-function for a given policy can be computed by iterating the Bellman operator $T^\pi$, defined by $T^\pi Q(s, a) = r(s, a) + \gamma P^\pi Q(s, a)$, where $P^\pi$ is the policy transition operator defined by $P^\pi Q(s, a) = \mathbb{E}_{s' \sim P(\cdot \mid s, a), a' \sim \pi(\cdot \mid s')}[Q(s', a')]$.

In offline RL, the agent is given a static dataset $D = \{(s, a, r, s')_i\}_{i=1}^N$, where the transitions are collected from the trajectories of an unknown behavior policy $\beta(a \mid s)$ interacting in the MDP. We denote $d_\beta(s)$ the state visitation frequency of $\beta$; the state-action pairs $(s, a)$ from the dataset $D$ are sampled from $d_\beta(s) \cdot \beta(a \mid s)$. Standard (offline) actor-critic methods alternate between evaluating the action value function of the current policy, and improving the current policy using the action value function. Specifically, at iteration $k$, given the current policy $\pi^{(k)}$ and the action value function $Q^{(k)}$, we perform policy evaluation and policy improvement as follows:

$$Q^{(k+1)} = \arg\min_Q \ \mathbb{E}_{(s,a,s') \sim D} \left[ Q(s, a) - \left( r(s, a) + \gamma \mathbb{E}_{a' \sim \pi^{(k)}(\cdot \mid s')}[Q^{(k)}(s', a')] \right) \right]^2 \quad \text{(policy evaluation)} \quad (1)$$

$$\pi^{(k+1)} = \arg\max_\pi \ \mathbb{E}_{s \sim D, a \sim \pi(\cdot \mid s)}[Q^{(k+1)}(s, a)] \quad \text{(policy improvement)} \quad (2)$$

Offline RL methods that perform vanilla policy evaluation and improvement based on these steps suffer from distribution shift during training, due to the evaluation of the Q-function on out-of-distribution (OOD) actions in the Bellman error. Specifically, the target Q-function $Q^{(k)}$ is evaluated on samples from the current policy $\pi^{(k)}$ to construct the Bellman target, i.e. $a' \sim \pi^{(k)}(\cdot \mid s')$, but the dataset contains only actions sampled from the behavior policy, i.e. $a' \sim \beta(\cdot \mid s')$, making its accuracy depend on Q-value estimates for actions outside the training distribution. This discrepancy between $\pi$ and $\beta$ can cause erroneous target Q-values, and overestimation occurs when the policy improvement step seeks to find a policy that maximizes $\mathbb{E}_{s \sim D, a \sim \pi(\cdot \mid s)}[Q(s, a)]$ (Kumar et al., 2019; Levine et al., 2020). This issues motivates the need for regularized objectives and conservative estimates of the value function.

## 4 Density-Scaled Regularization

We describe our Density-Scaled (DS) method for regularizing the policy evaluation step. We first motivate our work by discussing relevant existing forms of regularization and potential areas for improvement. Then, we construct our DS penalty and highlight its main theoretical results and appealing properties. Finally, we provide details for a practical implementation with Q-learning.

### 4.1 Motivation

We first describe how recent methods address the OOD and overestimation problem by regularizing the policy evaluation step. To learn the Q-function offline using samples from the behavior policy, we consider

the squared Bellman error $B(Q) := \frac{1}{2}\mathbb{E}_{s\sim D, a\sim\beta(\cdot|s)}[(Q(s,a) - T^\pi Q'(s,a))^2]$, where $Q'$ is a target copy of $Q$, treated as a constant in optimization, and $\pi$ is the current policy, which is the policy to be evaluated (we leave the functional dependence on $\pi$ implicit). We consider value regularization methods that augment the Bellman error with a regularization term R, leading to the optimization problem

$$\arg\min_Q B(Q) + \alpha R(Q) \tag{3}$$

where $\alpha$ controls the strength of regularization. The regularization term is chosen in such a way so as to penalize the Q-function to avoid overestimation from OOD actions. Different choices of R lead to quantitative differences in the solutions to the regularized policy update (3). Here, we investigate the limitations of recent methods, highlighting areas where the behavior of their solutions may be sub-optimal. In particular, we consider the regularization term for two recent methods, CQL (Kumar et al., 2020) and SVR (Mao et al., 2023) and compare their corresponding solutions.

In CQL, the regularization term is given by

$$R_{\mathrm{CQL}}(Q) = \mathbb{E}_{s\sim D, a\sim\pi(\cdot|s)}[Q(s,a)] - \mathbb{E}_{s\sim D, a\sim\beta(\cdot|s)}[Q(s,a)], \tag{4}$$

and an optimal solution to the regularized evaluation problem in Equation (3) is given by

$$Q_{\mathrm{CQL}}(s,a) = \begin{cases} T^\pi Q(s,a) - \alpha\left(\frac{\pi(a|s)}{\beta(a|s)} - 1\right) & \beta(a\mid s) > 0 \\ -\infty & \beta(a\mid s) = 0, \pi(a\mid s) > 0 \\ Q(s,a) & \beta(a\mid s) = \pi(a\mid s) = 0 \end{cases}$$

for all $a \in A$ and $s \in D$. In SVR, the regularization term is given by

$$R_{\mathrm{SVR}}(Q) = \mathbb{E}_{s\sim D}\left[\sum_{a\notin\mathrm{supp}(\beta)} \mu(a\mid s)(Q(s,a) - Q_{\min})^2\right] \tag{5}$$

$$= \mathbb{E}_{s\sim D, a\sim\mu(\cdot|s)}[(Q(s,a) - Q_{\min})^2] - \mathbb{E}_{s\sim D, a\sim\beta(\cdot|s)}\left[\frac{\mu(a\mid s)}{\beta(a\mid s)}(Q(s,a) - Q_{\min})^2\right] \tag{6}$$

with $Q_{\min}$ being the minimum Q-value (for any policy) in the dataset and $\mu$ is a policy with a wider support than $\beta$, set to be the current policy with added Gaussian noise. The optimal solution to the regularized evaluation problem in Equation (3) is given by (Mao et al., 2023, Eq. (7))

$$Q_{\mathrm{SVR}}(s,a) = \begin{cases} T^\pi Q(s,a) & \beta(a\mid s) > 0 \\ Q_{\min} & \beta(a\mid s) = 0 \end{cases}$$

for all $a \in A$ and $s \in D$. We provide formal statements and expanded proofs of the above solutions in Appendix A.2.

While CQL smoothly reduces the Q-function by a term that depends on the ratio $\pi(a\mid s)/\beta(a\mid s)$, it is not lower-bounded, and can overly penalize the Q-function even for in-distribution actions. On the other hand, the solution provided by SVR is discontinuous, entirely avoiding penalization of in-distribution values. Although the SVR solution ensures that in-distribution Q-values are not penalized, it does not consider the possibility that in-distribution regions can still lead to poor estimates of the value function, especially in cases where the action density of the behavior policy is low. We are thus motivated to reconcile the positive aspects of both methods, seeking a regularization term that can smoothly vary the level of penalization based on the action density, while avoiding over-conservatism. To this end, we introduce our Density-Scaled (DS) penalty.

## 4.2 Density-Scaled Regularization

Based on the previous considerations, we construct a regularization term that achieves two properties: (1) underestimates the Q-function proportionately to the behavior density, which is a natural measure for how

'in-distribution' an action is; and (2) lower bounds the Q-function on out-of-support actions. Our DS method uses the following penalty:

$$R_{DS}(Q) = \mathbb{E}_{s \sim D, a \sim \pi(\cdot|s)} \left[ \left( 1 - \frac{\beta(a \mid s)}{\sup_{a'} \beta(a' \mid s)} \right) (Q(s,a) - Q_{\min})^2 \right]. \tag{7}$$

The regularization term involves an expectation over the policy $\pi$, which may be the current policy or a policy chosen to weight the penalty term for actions deemed important. In general, to penalize OOD actions effectively, $\pi$ should be chosen to have greater support or a wider distribution than $\beta$.

In practice, computing the penalty requires an explicit estimate $\hat{\beta}$ of the behavior density. A smooth estimator enables us to interpolate the action density from the dataset $D$ and query it on arbitrary (including OOD) actions outside the dataset. Here, we learn a smooth behavior policy $\hat{\beta}(a|s; \psi)$ parameterized by $\psi$ using neural networks, which are powerful interpolators.

We consider policy evaluation with this penalty and characterize its solution in Theorem 1.

**Theorem 1** (DS update). *Given a policy $\pi(a \mid s)$, behavior policy $\beta(a \mid s)$, and regularization parameter $\alpha$, let $C_s = \sup_{a'} \beta(a' \mid s)$ for each $s$, and $k_{s,a} = \left( \frac{\pi(a|s)}{\beta(a|s)} - \frac{\pi(a|s)}{C_s} \right)$. For $a \in A$ and $s \in D$, the solution to*

$$\arg\min_Q B(Q) + \frac{\alpha}{2} \mathbb{E}_{s \sim D, a \sim \pi(\cdot|s)} \left[ \left( 1 - \frac{\beta(a \mid s)}{C_s} \right) (Q(s,a) - Q_{\min})^2 \right] \tag{8}$$

*is given by*

$$Q_{DS}(s,a) = \begin{cases} T^\pi Q(s,a) & \beta(a \mid s) = C_s \\ \frac{1}{1+\alpha k_{s,a}} T^\pi Q(s,a) + \frac{\alpha k_{s,a}}{1+\alpha k_{s,a}} Q_{\min} & 0 < \beta(a \mid s) < C_s \\ Q_{\min} & \beta(a \mid s) = 0 \end{cases} \tag{9}$$

All proofs are in Appendix A.2. Inspecting the solution, we see that the DS objective results in the standard policy target $T^\pi Q(s,a)$ when the action $a$ is at the mode of the behavior distribution. When $0 < \beta(a \mid s) < C_s$, the solution is given by a convex combination between $T^\pi Q(s,a)$ and $Q_{\min}$. On OOD actions, the Q-function is maximally penalized to $Q_{\min}$. An interesting property is that it can be viewed as a generalization, or smooth interpolation, of SVR.

**Observation 1.** $Q_{DS}$ *tends to* $Q_{SVR}$ *as* $\alpha \to 0$.

This provides another way to compute an approximate support-constraint solution. Compared to SVR's regularization term (Equation equation 6), our DS penalty avoids computation of the importance ratio $\mu(a \mid s)/\beta(a \mid s)$, which can potentially have high variance and suffer from numerical instability. This behavior is investigated and empirically compared in Section 5.2.

We now consider the properties of the operator defined by the DS solution of Theorem 1. Specifically, given a policy $\pi$, we define the operator $T_{DS}^\pi$ by $T_{DS}^\pi Q(s,a) := Q_{DS}(s,a)$. In Proposition 1, we show that the operator preserves the contraction property, similar to the standard Bellman operator. We also demonstrate in Proposition 2 that repeated evaluations lead to a fixed point that satisfies a lower bound of the true value function.

**Proposition 1** (Contraction). *The operator $T_{DS}^\pi Q(s,a)$ is a $\gamma$-contraction operator in the $L_\infty$ norm.*

**Proposition 2** (Fixed point). *For any $\pi$, the fixed point of $T_{DS}^\pi$, denoted by $f$, exists and satisfies*

$$\begin{cases} Q_{\min} \leq f(s,a) \leq Q^\pi(s,a) & \beta(a \mid s) > 0 \\ f(s,a) = Q_{\min} & \beta(a \mid s) = 0 \end{cases}$$

These results ensure that policy evaluation with the DS penalty results in conservative estimates of the Q-function, while being lower bounded by the minimum Q value on out-of-support regions.

### 4.3 Practical Implementation

We now describe a practical implementation of our method, based on a conventional deep Q-learning algorithm (e.g. Mnih et al. (2013); Haarnoja et al. (2018)). It proceeds by first estimating the behavior policy from the dataset. Next, we perform approximate policy evaluation and policy improvement in alternating steps, where the objective for the Q-function is augmented with the DS penalty. The full algorithm is provided in Algorithm 1.

---

**Algorithm 1** Density-Scaled Q-learning

---

**Require:** Offline dataset $D = \{(s, a, r, s')\}$, DS regularization coefficient $\alpha$, target averaging rate $\tau$
 Initialize Q-network $Q(s, a; \theta)$, target Q-network $Q'(s, a; \theta')$, behavior policy $\beta(a \mid s; \psi)$, policy network $\pi(a \mid s; \phi)$
 ▷ Estimate behavior policy via behavior cloning
 **for** gradient_step $= 1$ to $M$ **do**
  Sample random minibatch $(s, a) \sim D$
  Update behavior policy parameters $\psi$ by minimizing Eq. (10)
 **end for**
 ▷ Policy and critic training
 **for** gradient_step $= 1$ to $N$ **do**
  Sample random minibatch of $(s, a, r, s') \sim D$
  Update critic parameters $\theta$ by minimizing Eq. (11)
  Update policy parameters $\phi$ by minimizing Eq. (12)
  Update target network parameters: $\theta' \leftarrow (1 - \tau)\theta' + \tau\theta$
 **end for**

---

**Behavior policy.** Given the dataset $D = \{(s, a, r, s')\}$, we estimate the behavior policy $\beta$ by learning a parameterized model $\beta(a \mid s; \psi)$ by maximum likelihood estimation, using the following loss:

$$\arg \min_{\psi} L_{\beta}(\psi) = -\mathbb{E}_{(s,a) \sim D} \log[\beta(a \mid s; \psi)]. \tag{10}$$

We use a multivariate Gaussian model for the behavior policy: $\beta(a \mid s; \psi) = N(a; \mu(s; \psi), \sigma^2(s; \psi))$.

**Policy evaluation.** Given the current policy $\pi(a \mid s; \phi_k)$, the estimated behavior policy $\beta(a \mid s; \psi)$, the current Q-network $Q(s, a; \theta)$, target Q-network $Q'(s, a; \theta')$, and a noise standard deviation $\sigma$, we update the parameters of the critic by minimizing the following loss:

$$\theta_{k+1} \leftarrow \arg \min_{\theta} L_Q(\theta) = \mathbb{E}_{(s,a,s') \sim D}[(Q(s, a; \theta) - r(s, a) - \gamma \mathbb{E}_{a' \sim \pi(\cdot|s';\phi_k)} Q'(s', a'; \theta'_k))^2]$$
$$+ \alpha \mathbb{E}_{s \sim D, a \sim \pi_{\sigma}(\cdot|s;\psi)} \left[ \left(1 - \frac{\beta(a \mid s; \phi_k)}{C_s}\right)(Q(s, a; \theta) - Q_{\min})^2 \right]. \tag{11}$$

When $\beta$ is the Gaussian model, we have $C_s = (\sigma(s)\sqrt{2\pi})^{-1}$. We set $Q_{\min} = r_{\min}/(1 - \gamma)$, where $r_{\min}$ is the minimum reward in the dataset. Similar to SVR, the sampling distribution in the penalty $\pi_{\sigma}(\cdot|s, \phi_k)$ is the action distribution induced by sampling $\tilde{a} \sim \pi(\cdot \mid s; \phi_k)$ and returning $a = \tilde{a} + \epsilon$, where $\epsilon \sim N(0, \sigma^2 I)$. The noise scale $\sigma$ is selected to increase the frequency of OOD action samples. The target network parameters $\theta'_k$ are updated via Polyak averaging, that is, $\theta'_k \leftarrow \tau\theta_k + (1 - \tau)\theta'_k$ where $\tau \in (0, 1)$ is the interpolation factor.

**Policy improvement.** Using the current estimate of the Q-function $Q(s, a; \theta)$, we update the policy parameters $\phi$ by minimizing the following loss:

$$\phi_{k+1} \leftarrow \arg \min_{\phi} L_{\pi}(\phi) = -\mathbb{E}_{(s,a) \sim D, a' \sim \pi(\cdot|s;\phi_k)}[Q(s, a'; \theta_{k+1}) - \lambda \| a - a' \|^2], \tag{12}$$

which follows the standard policy update loss with a behavior cloning term $\lambda \| a - a' \|^2$ (Tarasov et al., 2023) which we turn on only for the Adroit and Vision D4RL datasets in our experiments.

# 5 Experiments

We evaluate the effectiveness of our DS algorithm and compare it to previous offline RL methods including the state-of-the-art SVR method. We benchmark our method on offline continuous control problems that contain both vector observations and high-dimensional pixel observations. We also investigate and compare the robustness and numerical stability of our method to SVR by designing an experiment that modifies the behavior policy estimator. Finally, we study the various components of our method, including sensitivity to the hyperparameter values and different behavior density model classes.

## 5.1 Continuous control

**D4RL.** First, we benchmark a suite of continuous control tasks in D4RL (Fu et al., 2021). We compare our DS method with previous model-free baseline algorithms: BC, One-Step RL (Brandfonbrener et al., 2021), TD3+BC (Fujimoto & Gu, 2021), CQL (Kumar et al., 2020), IQL (Kostrikov et al., 2021a), and SVR (Mao et al., 2023). The BC scores are derived from (Mao et al., 2023). We tune the regularization strength $\alpha$ based on the difficulty of the environments: we set a single parameter across all datasets containing expert trajectories and the Adroit Pen environments, and a different parameter for the rest of the datasets. Full experimental details, including training parameters and hyperparameter choices, are provided in Appendix A.1.

Table 1: Normalized scores on the D4RL benchmarks. m = medium, m-r = medium-replay, m-e = medium-expert, e = expert, r = random. Results for the baseline algorithms are taken from the original papers. Mean scores and standard deviations are computed over 5 seeds for our algorithm.

| Dataset | BC | OneStep | TD3BC | CQL | IQL | SVR | DS (Ours) |
|---|---|---|---|---|---|---|---|
| halfcheetah-m | 42 | 50.4 | 48.3 | 47 | 47.4 | 60.5 | 62.8±0.4 |
| hopper-m | 56.2 | 87.5 | 59.3 | 53 | 66.2 | 103.5 | 104.0±0.6 |
| walker2d-m | 71 | 84.8 | 83.7 | 73.3 | 78.3 | 92.4 | 91.3±2.1 |
| halfcheetah-m-r | 36.4 | 42.7 | 44.6 | 45.5 | 44.2 | 52.5 | 54.5±0.6 |
| hopper-m-r | 21.8 | 98.5 | 60.9 | 88.7 | 94.7 | 103.7 | 103.4±0.3 |
| walker2d-m-r | 24.9 | 61.7 | 81.8 | 81.8 | 73.8 | 95.6 | 96.0±2.0 |
| halfcheetah-m-e | 59.6 | 75.1 | 90.7 | 75.6 | 86.7 | 94.2 | 94.2±2.5 |
| hopper-m-e | 51.7 | 108.6 | 98 | 105.6 | 91.5 | 111.2 | 110.4±1.3 |
| walker2d-m-e | 101.2 | 111.3 | 110.1 | 107.9 | 109.6 | 109.3 | 109.5±0.2 |
| halfcheetah-e | 92.9 | 88.2 | 96.7 | 96.3 | 95 | 96.1 | 94.4±0.8 |
| hopper-e | 110.9 | 106.9 | 107.8 | 96.5 | 109.4 | 111.1 | 111.2±0.5 |
| walker2d-e | 107.7 | 110.7 | 110.2 | 108.5 | 109.9 | 110 | 111.7±0.8 |
| halfcheetah-r | 2.6 | 2.3 | 11 | 17.5 | 13.1 | 27.2 | 26.6±1.0 |
| hopper-r | 4.1 | 5.6 | 8.5 | 7.9 | 7.9 | 31.0 | 31.1±0.2 |
| walker2d-r | 1.2 | 6.9 | 1.6 | 5.1 | 5.4 | 2.2 | 2.2±0.8 |
| gym-v2 total | 784.2 | 1041.2 | 1013.2 | 1010.2 | 1033.1 | 1200.5 | 1203.2±4.5 |
| pen-expert | 85.1 | 61.6 | 111 | 107 | 110.2 | 138.9 | 151.0±3.9 |
| pen-human | 34.4 | 73.7 | 54.9 | 37.5 | 71.5 | 73.1 | 52.4±12.9 |
| pen-cloned | 56.9 | 31.8 | 63.8 | 39.2 | 37.3 | 70.2 | 39.7±8.1 |
| adroit-v0 total | 176.4 | 167.1 | 229.7 | 183.7 | 219 | 282.2 | 243.0±15.7 |

Table 1 summarizes the average scores for each algorithm. In the Gym-Mujoco domains, DS is competitive with the state-of-the-art SVR across all tasks, slightly outperforming it on the total aggregate score. In the Adroit domain, DS achieves the highest performance on the expert domain, but underperforms SVR in the human and cloned datasets. We note that SVR uses additional hyperparameters specific for the pen-human and pen-cloned environments, where their regularization strength is set to 10, two orders of magnitude higher than their parameter for pen-expert (0.02). In contrast, the DS regularization strength is fixed for all pen environments, which, along with a standard actor penalty term, achieves good performance without needing to greatly vary the regularization strength for specific environments. Overall, this demonstrates that the DS penalty is competitive with state-of-the-art methods and comparable in performance to SVR.

**Vision D4RL.** We also evaluate our algorithm on challenging visual domains using the V-D4RL dataset (Lu et al., 2023), which is a pixel-based analogue of D4RL. We compare it against the results of the model-free algorithms investigated in the original paper and also our re-implementation of SVR for this problem. We run our algorithm with the actor regularization term (Eq. (12), $\lambda > 0$). For fairness, we also compare against SVR with the actor regularization term added; our experiments running the base SVR show that it fails to learn without our additional actor penalty. Experimental details are provided in Appendix A.1.

Table 2: Normalized scores on the V-D4RL benchmarks. Results for the baseline algorithms are taken from (Lu et al., 2023). Mean scores and standard deviations are computed from 3 seeds and are mapped from [0,1000] to [0,100].

| Environment | DrQ+BC | CQL | BC | SVR | DS (Ours) |
|---|---|---|---|---|---|
| walker-walk-random | 5.5 | 14.4 | 2.0 | 2.4±0.4 | 3.5±0.1 |
| walker-walk-mixed | 28.7 | 11.4 | 16.5 | 18.3±3.5 | 19.3±2.8 |
| walker-walk-medium | 46.8 | 14.8 | 40.9 | 42.2±3.4 | 40.9±1.8 |
| walker-walk-medexp | 86.4 | 56.4 | 47.7 | 45.0±8.4 | 45.9±10.2 |
| walker-walk-expert | 68.4 | 89.6 | 91.5 | 81.2±11.4 | 81.9±5.5 |
| cheetah-run-random | 5.8 | 5.9 | 0.0 | 0.0±0.0 | 0.7±0.1 |
| cheetah-run-mixed | 44.8 | 10.7 | 25 | 29.1±4.1 | 23.7±0.8 |
| cheetah-run-medium | 53.0 | 40.9 | 51.6 | 46.8±2.8 | 46.1±6.0 |
| cheetah-run-medexp | 50.6 | 20.9 | 57.5 | 51.2±4.2 | 53.8±12.2 |
| cheetah-run-expert | 34.5 | 61.5 | 67.4 | 41.1±13.0 | 42.4±6.6 |
| humanoid-walk-random | 0.1 | 0.2 | 0.1 | 0.1±0.0 | 0.1±0.0 |
| humanoid-walk-mixed | 15.9 | 0.1 | 18.8 | 13.9±2.9 | 14.5±5.5 |
| humanoid-walk-medium | 6.2 | 0.1 | 13.5 | 7.8±2.4 | 9.6±1.9 |
| humanoid-walk-medexp | 7.0 | 0.1 | 17.2 | 7.2±3.2 | 13.0±5.6 |
| humanoid-walk-expert | 2.7 | 1.6 | 6.1 | 3.3±0.4 | 3.5±0.7 |
| total | 300.6 | 328.6 | 455.8 | 389.6±21.5 | 399.0±21.0 |

Table 2 provides the full experimental results. Overall, we find that our method is competitive with existing methods, slightly outperforming DRQ+BC and CQL, and only underperforming the BC method. We note that Lu et al. (2023) finds that V-D4RL is challenging for model-free actor critic algorithms, with BC outperforming these methods on most tasks. However, when taking both datasets (standard D4RL and V-D4RL) into consideration, these findings demonstrate that our method can achieve good performance consistently across both standard offline RL tasks and tasks involving complex visual observations.

## 5.2 Robustness comparison with SVR

The preceding experiments suggest that our method and SVR performs similarly. We perform additional experiments to demonstrate that this is only true when the behavior policy uses a tuned standard deviation $\sigma$, which was set to 0.2 following the SVR paper: with an alternative choice of a fixed $\sigma$, our method is more stable across a range of values and significantly outperforms SVR when $\sigma$ is small; importantly, when $\sigma$ is learned, our method's performance remains strong, but SVR can fail badly. This is explained in detail below.

We note that both the DS method and SVR require an explicit density estimator $\hat{\beta}(a \mid s)$ of the behavior policy to compute its regularization term. SVR computes an importance ratio $\pi(a \mid s)/\hat{\beta}(a \mid s)$, whereas we use the scaling term $(1 - \hat{\beta}(a \mid s)/\sup_{a'} \hat{\beta}(a' \mid s))$. This difference can result in different optimization behaviors, since SVR's importance ratio is potentially unbounded, whereas the DS term is bounded in $[0, 1]$. Thus, we conduct experiments to investigate the robustness and numerical stability of each method for different behavior policies. This is done in two experiments:

(1) We artificially vary the spread of the learned behavior policy to test misestimation.

(2) We then learn and evaluate a behavior policy given by a Gaussian variance network that is more expressive than that which was used in the preceding benchmark experiments.

For both DS and SVR, the results of the benchmark experiments (Section 5.1) were generated by learning a parameterized Gaussian model $\hat{\beta}(a \mid s; \psi) = N(a; \mu(s; \psi), \sigma^2 I)$ for the behavior policy, where the standard deviation was set to be a constant $\sigma = 0.2$ (as in the original SVR paper).

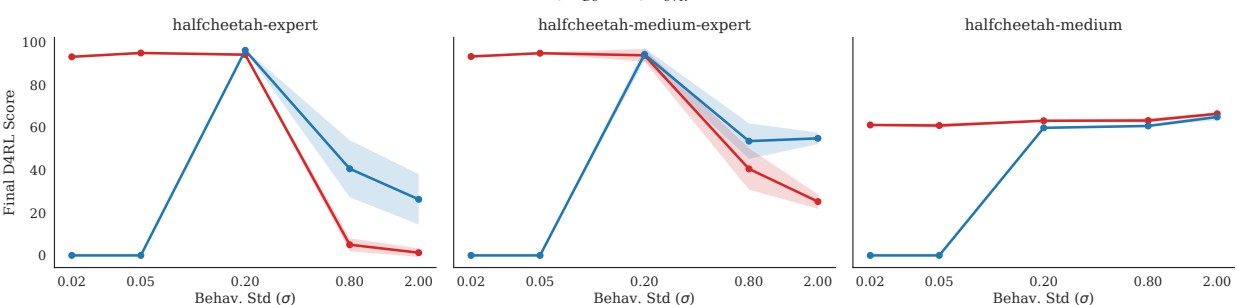

Figure 1: Results of varying the standard deviation parameter in the Gaussian behavior model $\beta$. Curves are averaged over 3 independent trials, where shading represents the standard deviation.

**Misestimated variance.** In the first experiment, we vary the $\sigma$ parameter across a range of values and evaluate the algorithms on these different behavior policies. The results are displayed in Figure 1. We discover that for small values of $\sigma = \{0.02, 0.05\}$, SVR fails to train as the penalty term diverges, leading to numerical overflow during optimization. As alluded to in the beginning, this issue arises because of the importance weight used in SVR regularization term, which is unbounded for arbitrary densities and blows up for any action $a$ such that $\hat{\beta}(a \mid s)$ is close to zero, which can occur in practice when the Gaussian density is narrow. In contrast, the bounded DS penalty results in superior numerical stability because it is bounded in $[0, 1]$, and our method outperforms or remaining close to SVR on most of the different $\sigma$ values.

**Expressive variance network.** In the second experiment, we consider a more practical scenario where the density estimator is a more expressive model, given by a Gaussian variance network that learns a state-dependent $\sigma(s)$. This reflects a realistic scenario where one aims to estimate the behavior policy as well as possible using maximally expressive models. We learn the parameters of this policy by minimizing the weighted negative log-likelihood (from Seitzer et al. (2022)):

$$L_{\mu,\sigma}(\psi) = \mathbb{E}_{(s,a)\sim D} \left[ \lfloor \sigma(s; \psi) \rfloor \left( \log \sigma^2(s; \psi) + \frac{\|\mu(s; \psi) - a\|^2}{\sigma^2(s; \psi)} \right) \right] \tag{13}$$

where $\lfloor \cdot \rfloor$ denotes the stop-gradient operation. This produces a behavior policy $\hat{\beta}(a \mid s) \sim N(\mu(s; \psi), \sigma^2(s; \psi))$ where the variance is a learned function of $s$, instead of being constant. Again, we train and evaluate the performance of SVR and DS on Gym-Mujoco using this behavior model. We present the aggregated results and comparison in Table 3, and provide the full results and training details in Appendix A.3.

| Dataset | DS | DS-var | SVR-var |
|---|---|---|---|
| halfcheetah-v2 total | 332.3 | 322.1 | Null |
| hopper-v2 total | 459.4 | 450.4 | Null |
| walker-v2 total | 409.9 | 404.1 | Null |

Table 3: Total scores on the Gym-Mujoco domain, summed over {random, medium, medium-replay, medium-expert, expert} datasets. '-var' refers to using the behavior model with a learnable variance. Null results arise from failure to train the algorithm.

| Environment | $\sigma = 0.2$ | $\sigma(s)$ |
|---|---|---|
| halfcheetah-expert-v2 | -2.2 | -3.3 |
| halfcheetah-medium-expert-v2 | -1.9 | -2.5 |
| halfcheetah-medium-v2 | -2.1 | -3.4 |

Table 4: Negative log-likelihoods of the behavior policy with fixed variance that was used to generate the results in Tables 1 and 3, and the policy with learned $\sigma(s)$.

Here, we find that SVR fails to train for all environments, due to the same issue of the penalty diverging early in training due to small values of the behavior density. As a sanity check, we compare the negative

log-likelihood (NLL) of the learned $\sigma$ with that of the tuned parameter $\sigma = 0.2$ in Table 4. As expected, the NLL for the learned $\sigma$ is lower than that for $\sigma = 0.2$ for all three benchmark environments. This indicates that while in practice, we often want to use a more expressive behavior model with a learnable variance to obtain a better fit to the observed experiences, this natural approach does not work well with SVR, as the learned $\sigma$ can be very small, leading to small behavior density and numerical issues in SVR. In contrast, our method work well with this approach. Hence, on new problems, SVR may require manually tuning the behavior policy (specifically, tuning the $\sigma$ parameter) to achieve a stable regularization, while DS admit the flexibility of estimating the behavior policy directly without needing to tune extra parameters. This makes DS easier to apply than SVR in practice.

### 5.3 Hyperparameter sensitivity and ablation studies

We perform a number of experiments to investigate the hyperparameter sensitivity, the effect of varying the sampling distribution of the penalty, and the performance with different behavior density model classes.

**Regularization strength.**     We evaluate the sensitivity of the DS penalty to the regularization strength $\alpha$. We focus on datasets of varying difficulty, ranging from medium to expert, and consider a range of values $\alpha \in [10^{-4}, 10^{-3}, ..., 1]$. The performance of our method is displayed in Figure 2. We observe that for datasets with expert or mixed expert trajectories, there is a range spanning an order of magnitude of $\alpha$ where the DS penalty performs well. For suboptimal datasets, where coverage is more uniform, the DS penalty and performs well on a wide range of magnitudes. This indicates that the performance and strength of regularization is sensitive to the coverage and optimality of the dataset, yet there is a reasonable range of values that can perform well across different tasks of similar dataset type.

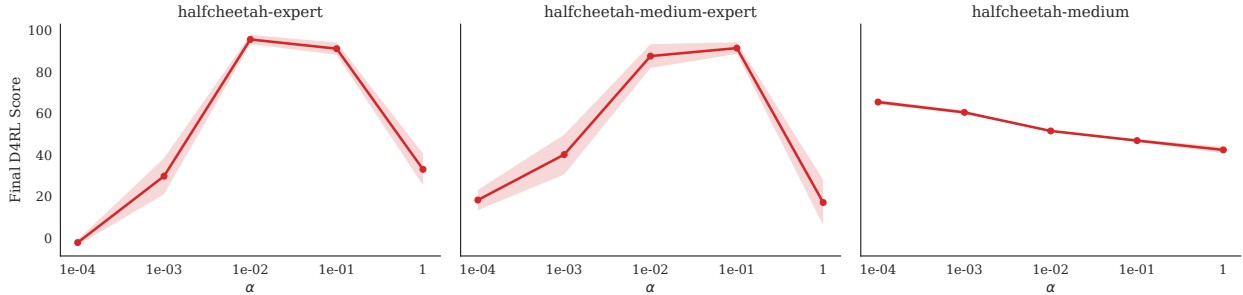

Figure 2: Results of varying the regularization parameter $\alpha$ for the DS penalty. Curves are averaged over 3 independent trials, where shading represents the standard deviation.

**Sampling distribution.**     We also vary the spread of the sampling distribution $\pi_\sigma(\cdot \mid s; \phi_k)$ (defined in Section 4.3) by modifying the noise scale $\sigma$ which determines how much noise is added to the action samples from the current policy. Figure 3 displays the performance of our method across a range of standard deviations. The results indicate that DS performs well in a wide range of values, including the noiseless case ($\sigma = 0$) where we simply use the current policy to sample actions, and higher values that approximate uniform sampling. Note that in the purely expert environments, adding too much noise (e.g. $\sigma = 0.8$) can substantially degrade the performance.

**Mixture density estimator.**     Our previous experiments used a Gaussian policy, either with fixed or learned variance, which performs well but is generally limited to learning unimodal behavior distributions. As an additional ablation, we replace this with a Gaussian mixture model (GMM) consisting of 3 components with learned diagonal covariance matrices, which defines a behavior density $\hat{\beta}(a \mid s) = \sum_{k=1}^{3} \gamma_k(s) N(a; \mu_k(s), \sigma_k^2(s))$. We approximate the global maximum of the density for each state by $\tilde{C}_s = \sum_{k=1}^{3} \gamma_k(s) \sup_a N(a; \mu_k(s), \sigma_k^2(s))$ which satisfies $0 \le \hat{\beta}(a \mid s)/\tilde{C}_s \le \hat{\beta}(a \mid s)/ \sup_{a'} \hat{\beta}(a' \mid s)$. We compare this GMM with the density estimator on the Gym-Mujoco tasks. Full details are provided in Appendix A.1.

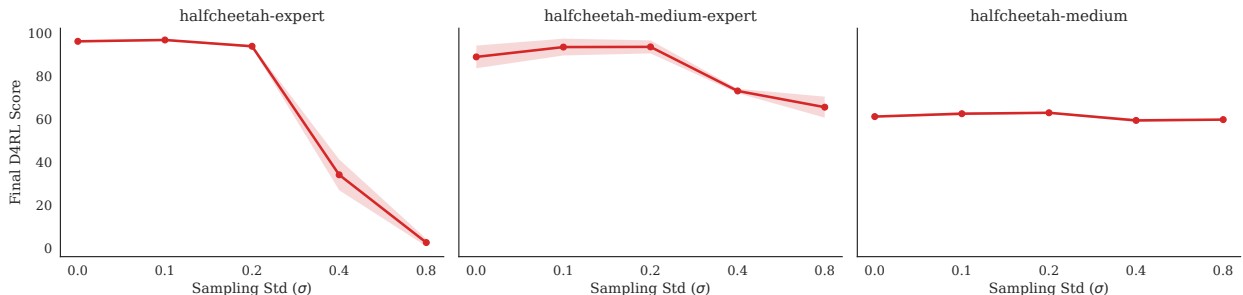

Figure 3: Results of varying the standard deviation of the sampling policy for the DS penalty. Curves are averaged over 3 independent trials, where shading represents the standard deviation.

The results are displayed in Table 5. We find that the mixture policy results in comparable performance, but the improved flexibility does not translate into improved downstream performance on datasets with mixed optimality. This may be because fitting the mixture model is more difficult in general and may have led to more estimation error, and we have incurred approximation error from our estimate of the global maximum of the density. Despite this, these results show that DS penalty can adapt to and perform well with more flexible density models that can model multi-modal behaviors.

Table 5: Comparison with mixture density estimator, denoted DS-mix. m-e = medium-expert. Results are averaged across 5 seeds. Total results are across all environments; individual results in Appendix A.3.

| Dataset | DS-var | DS-mix |
|---|---|---|
| halfcheetah-m-e | 94.2 | 94.5 |
| hopper-m-e | 110.4 | 103.9 |
| walker-m-e | 109.5 | 90.4 |
| gym-v2 total | 1203.2 | 1159.3 |

## 6 Conclusion

This work considers value-based offline RL and proposes a new regularization method, the Density-Scaled penalty, that reduces out-of-distribution errors by penalizing the policy evaluation step. The penalty is simple and practical to implement for value-based RL methods. We theoretically connect it to existing state-of-the-art work, SVR, and demonstrate that it is competitive on the benchmark datasets while being more numerically stable and robust to behavior density estimation.

We have demonstrated that the DS penalty provides a promising alternative for performing regularization on offline Q-learning algorithms. Like SVR, our method requires explicit density estimation of the behavior policy, and future work can investigate more refined models that could capture the dataset distribution more accurately or methods to compute the loss without explicitly constructing this estimator. Our experiments testing a flexible mixture density model provides preliminary evidence that our regularization term can be effective with multi-modal behavior densities. For datasets with stronger multimodality that require more flexible density models, computing the DS regularization term, which involves normalizing by the global maximum of the behavior density, is more challenging and generally intractable. An interesting direction for future work is to consider approximations to the penalty term that can be computed efficiently for general behavior policy models. In addition, our general idea of penalizing the Q-values in proportion to the behavior density could be extended by considering other forms beyond the quadratic, or setting a suitably defined state-action dependent minimum value. This could be useful for improving the performance and increasing the flexibility of the method.

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

# A Appendix

## A.1 Experimental details

In this section, we provide further details on the experiments conducted.

**D4RL.** Table 6 provides the hyperparameter choices for our DS method.

Table 6: Hyperparameters for implementation of DS algorithm for the D4RL datasets.

| Hyperparameter | Value |
|---|---|
| Actor hidden dimension | [256, 256] |
| Critic hidden dimension | [256, 256] |
| Number of critics | 4 |
| Actor learning rate | $3 \times 10^{-4}$ |
| Critic learning rate | $3 \times 10^{-4}$ |
| Optimizer | Adam |
| Critic penalty coefficient $\alpha$ | {0.0009, 0.04} |
| Actor penalty coefficient $\lambda$ | 1.0 (Adroit) |
| Target update rate $\tau$ | 0.005 |
| Batch size | 256 |
| Discount factor $\gamma$ | 0.99 |
| Number of iterations | $10^6$ |
| Policy update frequency | 2 |

For Gym-Mujoco datasets, we use a penalty coefficient $\alpha = 0.0009$ for random, medium, and medium-replay environments, and $\alpha = 0.04$ for medium-expert and expert environments. We do not use actor regularization for Gym-Mujoco (i.e. $\lambda = 0$). For the Adroit datasets, we use $\alpha = 0.04$ for all environments, with an actor penalty coefficient of $\lambda = 1.0$.

The behavior model is a network with the same architecture as the actor. It is trained with Adam (lr=0.001) for $10^5$ iterations. For the Gaussian model we used a fixed standard deviation of $\sigma = 0.2$.

**V-D4RL.** The implementation of DS uses the same critic and actor network and the same training hyperparameters as the CQL agent in the V-D4RL paper (Lu et al., 2023). We train both SVR and DS for $6 \times 10^5$ steps.

For our critic and actor regularization hyperparameters, we set $\alpha = 0.1$ and $\lambda = 1$ for all environments.

For the behavior network, we use the same architecture and training procedure as the BC agent in the V-D4RL paper. For the Gaussian model we used a fixed standard deviation of $\sigma = 0.1$.

**Variance network.** The behavior model has the same network backbone as the baseline Gaussian policy, with an extra final layer of size $[256, |A|]$ that outputs the diagonal variance terms Softplus$[\sigma_\psi^2(s)]$. It is trained with the same training hyperparameters as the baseline.

**GMM experiment.** For our experiment with the 3-component GMM behavior policy, we use a network with the same backbone as the baseline Gaussian policy, with three output layers that output the mixture logits, means, and diagonal variances. The model is trained by minimizing the negative log-likelihood:

$$\arg\min_{\psi} -\mathbb{E}_{(s,a)\sim D} \log \hat{\beta}(a|s;\psi) = -\mathbb{E}_{(s,a)\sim D} \log \left( \sum_{k=1}^{3} \gamma_k(s;\psi) N(a; \mu_k(s;\psi), \sigma_k^2(s;\psi)) \right), \qquad (14)$$

using the same training hyperparameters as for the baseline.

For evaluation, we use the same DS regularization strength as the baseline for all environments, and set the penalty sampling noise to be $\sigma = 0.1$.

## A.2 Proofs

In this section, we provide proofs of the main results in Section 4. We will at times use $\mathbb{E}_s$ as a shorthand for $\mathbb{E}_{s\sim D}$, $\mathbb{E}_{s'}$ for $\mathbb{E}_{s'\sim P(\cdot|s,a)}$, and $\mathbb{E}_{a\sim\beta}$ for $\mathbb{E}_{a\sim\beta(\cdot|s)}$ when the context is clear. We prove the main results regarding the update solution for the DS objective and its properties. We also restate and provide expanded proofs of the update solutions to the CQL and SVR objectives; the full proofs of the CQL and SVR solutions were not provided in the original papers.

We use the following fact about the derivative of the Bellman error. Given dataset $D = \{(s, a, s', r)\}$, where $a \sim \beta(\cdot \mid s)$ for each $s$, the mean squared Bellman error is given by

$$B(Q) = \frac{1}{2}\mathbb{E}_{s\sim D,a\sim\beta(\cdot|s)}[(Q(s,a) - T^\pi Q'(s,a))^2].$$

Note that here, we write $Q'$ (a target copy of $Q$) in the target $T^\pi Q'(s,a)$ since we are treating it as a fixed constant in optimization. The derivative of the error with respect to the current value $Q(s,a)$ is:

$$\left.\frac{dB(Q)}{dQ(s,a)}\right|_{Q'(s,a)=Q(s,a)} = d_\beta(s)\beta(a \mid s)\left[Q(s,a) - T^\pi Q(s,a)\right],$$

where $d_\beta(s)$ is the state-visitation distribution of $\beta(a \mid s)$. We assume throughout that $d_\beta(s) > 0$ for all $s$.

**Theorem 1** (DS update). *Given a policy $\pi(a \mid s)$, behavior policy $\beta(a \mid s)$, and regularization parameter $\alpha$, let $C_s = \sup_{a'} \beta(a' \mid s)$ for each $s$, and $k_{s,a} = \left(\frac{\pi(a|s)}{\beta(a|s)} - \frac{\pi(a|s)}{C_s}\right)$. For $a \in A$ and $s \in D$, the solution to*

$$\arg\min_Q B(Q) + \frac{\alpha}{2}\mathbb{E}_{s\sim D,a\sim\pi(\cdot|s)}\left[\left(1 - \frac{\beta(a \mid s)}{C_s}\right)(Q(s,a) - Q_{\min})^2\right] \tag{8}$$

*is given by*

$$Q_{\mathrm{DS}}(s,a) = \begin{cases} T^\pi Q(s,a) & \beta(a \mid s) = C_s \\ \frac{1}{1+\alpha k_{s,a}}T^\pi Q(s,a) + \frac{\alpha k_{s,a}}{1+\alpha k_{s,a}}Q_{\min} & 0 < \beta(a \mid s) < C_s \\ Q_{\min} & \beta(a \mid s) = 0 \end{cases} \tag{9}$$

*Proof.* For fixed $(s,a)$, consider the following cases. We assume that $\pi(a \mid s) > 0$ throughout.

Case 1: $\beta(a \mid s) = C_s$. In this case, we have $(1 - \frac{\beta(a|s)}{C_s}) = 0$, so the regularization term vanishes, and the solution is the same as the unregularized objective: $Q^*(s,a) = T^\pi Q(s,a)$.

Case 2: $0 < \beta(a \mid s) < C_s$. Setting the derivative of the objective with respect to $Q(s,a)$ to zero, we have

$$\frac{dB(Q)}{dQ(s,a)} + d_\beta(s)\alpha[\pi(a \mid s)\left(1 - \frac{\beta(a \mid s)}{C_s}\right)(Q(s,a) - Q_{\min})] = 0$$

$$\beta(s,a)[Q(s,a) - T^\pi Q(s,a)] + \alpha\,\pi(a \mid s)\left(1 - \frac{\beta(a \mid s)}{C_s}\right)(Q(s,a) - Q_{\min}) = 0$$

$$Q(s,a) - T^\pi Q(s,a) + \alpha\left(\frac{\pi(a \mid s)}{\beta(a \mid s)} - \frac{\pi(a \mid s)}{C_s}\right)(Q(s,a) - Q_{\min}) = 0$$

Let $k_{s,a} = \frac{\pi(a|s)}{\beta(a|s)} - \frac{\pi(a|s)}{C_s}$. Then the above simplifies to

$$(1 + \alpha k_{s,a})Q(s,a) = \alpha k_{s,a}Q_{\min} + T^\pi Q(s,a)$$

$$Q(s,a) = \frac{1}{1+\alpha k_{s,a}}T^\pi Q(s,a) + \frac{\alpha k_{s,a}}{1+\alpha k_{s,a}}Q_{\min}.$$

Thus $Q^*(s,a) = \frac{1}{1+\alpha k}T^\pi Q(s,a) + \frac{\alpha k_{s,a}}{1+\alpha k_{s,a}}Q_{\min}$.

Case 3: $\beta(a \mid s) = 0$. In this case, the terms in the objective involving expectations over $a \sim \beta$ vanish. The objective becomes

$$\min_Q \alpha\mathbb{E}_{s\sim D,a\sim\pi}(Q(s,a) - Q_{\min})^2$$

which has the solution $Q^*(s,a) = Q_{\min}$.

Combining the cases, we obtain the update solution for the DS objective:

$$Q_{\mathrm{DS}} = \begin{cases} T^\pi Q(s,a) & \beta(a \mid s) = C_s \\ \frac{1}{1+\alpha k_{s,a}} T^\pi Q(s,a) + \frac{\alpha k_{s,a}}{1+\alpha k_{s,a}} Q_{\min} & 0 < \beta(a \mid s) < C_s \\ Q_{\min}(s,a) & \beta(a \mid s) = 0 \end{cases}$$

$\square$

Now, we prove the main properties of the DS operator. For simplicity, we rewrite the operator in a simpler way. Let $k_{s,a} = \left( \frac{\pi(a|s)}{\beta(a|s)} - \frac{\pi(a|s)}{C_s} \right)$ and $C_s = \sup_{a'} \beta(a' \mid s)$. For simplicty, we can set $\delta = \frac{1}{1+\alpha k_{s,a}}$, leaving the dependence on $s, a$ implicit. We thus have $1 - \delta = \frac{\alpha k_{s,a}}{1+\alpha k_{s,a}}$, and $0 < \delta < 1$. Then the operator $T_{\mathrm{DS}}^\pi Q(s,a)$ can be rewritten as

$$T_{\mathrm{DS}}^\pi Q(s,a) = \begin{cases} T^\pi Q(s,a) & \beta(a \mid s) = C_s \\ \delta T^\pi Q(s,a) + (1 - \delta) Q_{\min} & 0 < \beta(a \mid s) < C_s \\ Q_{\min} & \beta(a \mid s) = 0 \end{cases}$$

**Proposition 1** (Contraction). *The operator $T_{\mathrm{DS}}^\pi Q(s,a)$ is a $\gamma$-contraction operator in the $L_\infty$ norm.*

*Proof.* Let $f_1$ and $f_2$ be two arbitrary functions on $S \times A$. We consider the three cases induced by $\beta(a \mid s)$.

Case 1: $\beta(a \mid s) = 0$. In this case,

$$|T_{\mathrm{DS}}^\pi f_1(s,a) - T_{\mathrm{DS}}^\pi f_2(s,a)| = | Q_{\min} - Q_{\min} | = 0 \le \gamma \mid f_1 - f_2 \mid_\infty .$$

Case 2: $0 < \beta(a \mid s) < C_s$. We have

$$\begin{aligned} |T_{\mathrm{DS}}^\pi f_1(s,a) - T_{\mathrm{DS}}^\pi f_2(s,a)| &= |\delta(T^\pi f_1(s,a) - T^\pi f_2(s,a))| \\ &= \delta\gamma \left| \mathbb{E}_{s' \sim P(\cdot|s,a), a' \sim \pi(\cdot|s)} [f_1(s',a') - f_2(s',a')] \right| \\ &\le \gamma \mathbb{E}_{s' \sim P(\cdot|s,a), a' \sim \pi(\cdot|s)} [|f_1(s',a') - f_2(s',a')|,] \\ &\le \gamma \mid f_1 - f_2 \mid_\infty . \end{aligned}$$

Case 3: $\beta(a \mid s) = 0$. In this case, $T_{\mathrm{DS}}^\pi$ coincides with the standard Bellman operator and we have

$$\begin{aligned} |T_{\mathrm{DS}}^\pi f_1(s,a) - T_{\mathrm{DS}}^\pi f_2(s,a)| &= |T^\pi f_1(s,a) - T^\pi f_2(s,a)| \\ &\le \gamma \|f_1 - f_2\|_\infty. \end{aligned}$$

$\square$

**Proposition 2** (Fixed point). *For any $\pi$, the fixed point of $T_{\mathrm{DS}}^\pi$, denoted by $f$, exists and satisfies*

$$\begin{cases} Q_{\min} \le f(s,a) \le Q^\pi(s,a) & \beta(a \mid s) > 0 \\ f(s,a) = Q_{\min} & \beta(a \mid s) = 0 \end{cases}$$

*Proof.* Our proof is partially adapted from the technique used in Mao et al. (2023). By Proposition 1 and the contraction mapping theorem there exists a unique fixed point, which we denote $f$. The fixed point satisfies $f(s,a) = T_{\mathrm{DS}}^\pi f(s,a)$. The equality $f(s,a) = Q_{\min}$ where $\beta(a|s) = 0$ follows directly from the definition of $T_{\mathrm{DS}}^\pi$.

For $\beta(a|s) > 0$, we can combine the first two cases of $T_{\mathrm{DS}}^\pi$ and write $T_{\mathrm{DS}}^\pi f(s,a) = \delta T^\pi f(s,a) + (1 - \delta) Q_{\min}$, where $0 < \delta \le 1$. Note that taking $\delta = 1$ recovers the first case where $\beta(a|s) = C_s$.

We first prove the lower bound $f(s,a) \ge Q_{\min}$ for all $(s,a)$.

Fix $(s, a)$ with $\beta(s, a) > 0$. We have

$$
\begin{aligned}
f(s, a) &= T_{\mathrm{DS}}^\pi f(s, a) \\
&= \delta T^\pi f(s, a) + (1 - \delta) Q_{\min} \\
&= \delta[r(s, a) + \gamma \mathbb{E}_{s'} \mathbb{E}_{a' \sim \pi(\cdot \mid s')} f(s', a')] + (1 - \delta) Q_{\min} \\
&= \delta[r(s, a) + \gamma \mathbb{E}_{s'} \mathbb{E}_{a' \sim \pi(\cdot \mid s')} T^\pi f(s', a')] + (1 - \delta) Q_{\min}
\end{aligned}
$$

We now analyze the term $\mathbb{E}_{s'} \mathbb{E}_{a' \sim \pi(\cdot \mid s')} T^\pi f(s', a')$. Denote $I(s) = \{a \in \mathcal{A} \mid \beta(a \mid s) > 0\}$, $J(s) = \{a \in \mathcal{A} \mid \beta(a \mid s) = 0\}$. The expectation over actions may be expanded as follows:

$$
\begin{aligned}
\mathbb{E}_{a' \sim \pi(\cdot \mid s')} T^\pi f(s', a') &= \sum_{a' \in I(s')} \pi(a' \mid s') T^\pi f(s', a') + \sum_{a' \in J(s')} \pi(a' \mid s') T^\pi f(s', a') \\
&= \sum_{a' \in I(s')} \pi(a' \mid s') f(s', a') + \sum_{a' \in J(s')} \pi(a' \mid s') Q_{\min} \\
&\geq \sum_{a' \in I(s')} \pi(a' \mid s') f_{\min}(s, a) + \sum_{a' \in J(s')} \pi(a' \mid s') Q_{\min}
\end{aligned}
$$

where $f_{\min} := \min_{s \in S, a \in I(s)} f(s, a)$. Combining this result with the previous result, we have

$$
\begin{aligned}
f(s, a) &= \delta[r(s, a) + \gamma \mathbb{E}_{s'} \mathbb{E}_{a' \sim \pi(\cdot \mid s')} T^\pi f(s', a')] + (1 - \delta) Q_{\min} \\
&\geq \delta \left[ r(s, a) + \gamma \mathbb{E}_{s'} \left[ \sum_{a' \in I(s')} \pi(a' \mid s') f_{\min}(s, a) + \sum_{a' \in J(s')} \pi(a' \mid s') Q_{\min} \right] \right] + (1 - \delta) Q_{\min} \\
&\geq \delta[r_{\min}(s, a) + \gamma \lambda f_{\min}(s, a) + \gamma(1 - \lambda) Q_{\min}]] + (1 - \delta) Q_{\min}
\end{aligned}
$$

where in the last line we have set $\lambda := \mathbb{E}_{s'} \left[ \sum_{a' \in I(s')} \pi(a' \mid s') \right]$ and used the fact that

$$
1 = \mathbb{E}_{s'} \sum_{a' \in I(s') \cup J(s')} \pi(a' \mid s') = \mathbb{E}_{s'} \sum_{a' \in I(s')} \pi(a' \mid s') + \sum_{a' \in J(s')} \pi(a' \mid s') = \lambda + (1 - \lambda).
$$

Note that the relation $f(s, a) \geq \delta[r_{\min}(s, a) + \gamma \lambda f_{\min}(s, a) + (1 - \lambda) Q_{\min}]] + (1 - \delta) Q_{\min}$ holds for all $(s, a)$ such that $\beta(a \mid s) > 0$ and therefore also for the case where the LHS attains the minimum $f_{\min}$. Thus we have

$$
\begin{aligned}
f_{\min}(s, a) &\geq \delta[r_{\min}(s, a) + \gamma \lambda f_{\min}(s, a) + (1 - \lambda) Q_{\min}]] + (1 - \delta) Q_{\min} \\
&= \frac{1}{1 - \delta \gamma \lambda} [\delta(1 - \gamma) Q_{\min} + \delta \gamma(1 - \lambda) Q_{\min} + (1 - \delta) Q_{\min}] \\
&= Q_{\min}
\end{aligned}
$$

where we have used the fact that $Q_{\min} = r_{\min}/(1 - \gamma)$.

Therefore, we have $f(s, a) \geq f_{\min}(s, a) \geq Q_{\min}$ for all $(s, a)$ such that $\beta(a \mid s) > 0$.

Now, we prove the upper bound $f(s, a) \leq Q^\pi(s, a)$. To this end, we first show that $T_{\mathrm{DS}}^\pi f(s, a) \leq T^\pi f(s, a)$. Since $T_{\mathrm{DS}}^\pi f(s, a) = \delta T^\pi f(s, a) + (1 - \delta) Q_{\min}$ and $\delta \in [0, 1]$ it suffices to show that $T^\pi f(s, a) \geq Q_{\min}$. From the definition of $T$, we have for any $(s, a)$,

$$
\begin{aligned}
T^\pi f(s, a) &= r(s, a) + \gamma \mathbb{E}_{s'} \mathbb{E}_{a' \sim \pi(\cdot \mid s')} f(s', a') \\
&\geq r_{\min} + \gamma \mathbb{E}_{s'} \mathbb{E}_{a' \sim \pi(\cdot \mid s')} Q_{\min} \\
&= (1 - \gamma) Q_{\min} + \gamma Q_{\min} \\
&= Q_{\min}
\end{aligned}
$$

we we have used the result that $f(s, a) \geq Q_{\min}$ for all $(s, a)$ in the second inequality. Thus, we have $T_{\mathrm{DS}}^\pi f(s, a) \leq T^\pi f(s, a)$ for all $(s, a)$. Then, by induction,

$$
f(s, a) = T_{\mathrm{DS}}^\pi f(s, a) \leq T^\pi f(s, a) = T^\pi(T_{\mathrm{DS}}^\pi f(s, a)) \leq T^\pi(T^\pi f(s, a)) \ldots \leq (T^\pi)^n f(s, a).
$$

Letting $n \to \infty$, and using the fact that $Q^\pi$ is the fixed point of $T^\pi$, we have $f(s, a) \leq Q^\pi(s, a)$ for all $(s, a)$.

$\square$

Next, we provide expanded proofs for the CQL and SVR update solutions.

**Proposition 3** (CQL solution). *For any two policies $\beta(a \mid s)$ and $\pi(a \mid s)$, the solution to*

$$\min_Q B(Q) + \alpha(\mathbb{E}_{s \sim D}[\mathbb{E}_{a \sim \pi(\cdot \mid s)}[Q(s, a)] - \mathbb{E}_{a \sim \beta(\cdot \mid s)}[Q(s, a)]])$$

*is given by*

$$Q_{\text{CQL}}(s, a) = \begin{cases} T^\pi Q(s, a) - \alpha \left( \frac{\pi(a\mid s)}{\beta(a\mid s)} - 1 \right) & \beta(a \mid s) > 0 \\ -\infty & \beta(a \mid s) = 0, \pi(a \mid s) > 0 \\ Q(s, a) & \beta(a \mid s) = \pi(a \mid s) = 0 \end{cases}$$

*where we (arbitrarily) set the solution in the case $\beta(a \mid s) = \pi(a \mid s) = 0$ to be $Q(s, a)$, i.e. no update performed.*

*Proof.* The CQL update is

$$\min_Q B(Q) + \alpha(\mathbb{E}_{s \sim D, a \sim \pi}[Q(s, a)] - \mathbb{E}_{s \sim D, a \sim \beta}[Q(s, a)])$$

and is convex in $Q$. For fixed $(s, a)$, consider the following cases:

Case 1: $\beta(a \mid s) > 0$. Setting the derivative with respect to $Q(s, a)$ to zero, and dropping the $d_\beta(s)$ terms, we have

$$\beta(a \mid s) [Q(s, a) - T^\pi Q(s, a)] + \alpha(\pi(a \mid s) - \beta(a \mid s)) = 0.$$

Solving for $Q$ gives

$$Q^*(s, a) = T^\pi Q(s, a) - \alpha \frac{\pi(a \mid s) - \beta(a \mid s)}{\beta(a \mid s)}$$

$$= T^\pi Q(s, a) - \alpha \left( \frac{\pi(a \mid s)}{\beta(a \mid s)} - 1 \right).$$

Case 2: $\beta(a \mid s) = 0$ and $\pi(a \mid s) > 0$. In this case, the terms involving expectations in $\beta$ vanish, and the objective reduces to

$$\min_Q \alpha \mathbb{E}_{s \sim D, a \sim \pi}[Q(s, a)]$$

Since $\alpha > 0$ and $\pi(a \mid s) > 0$, the objective is minimized at $Q^*(s, a) = -\infty$.

Case 3: $\beta(a \mid s) = 0$ and $\pi(a \mid s) = 0$. Both the Bellman error term and the regularization term vanish. In this case, the solution is degenerate so we say $Q(s, a)$ is left unchanged.

Combining all cases, the CQL minimizer is given by

$$Q_{\text{CQL}} = \begin{cases} T^\pi Q(s, a) - \alpha \left( \frac{\pi(a\mid s)}{\beta(a\mid s)} - 1 \right) & \beta(a \mid s) > 0 \\ -\infty & \beta(a \mid s) = 0, \pi(a \mid s) > 0 \\ Q(s, a) & \beta(a \mid s), \pi(a \mid s) = 0 \end{cases}$$

$\square$

**Proposition 4** (SVR update). *For any policies $\beta(a \mid s)$, $\mu(a \mid s)$, and $\pi(a \mid s)$, the solution to*

$$\min_Q B(Q) + \alpha \mathbb{E}_{s \sim D} \left( \mathbb{E}_{a \sim \mu(\cdot \mid s)}(Q(s, a) - Q_{\min})^2 - \mathbb{E}_{a \sim \beta(\cdot \mid s)} \frac{\mu(a \mid s)}{\beta(a \mid s)}(Q(s, a) - Q_{\min})^2 \right)$$

*is given by*

$$Q_{\text{SVR}}(s, a) = \begin{cases} T^\pi Q(s, a) & \beta(a \mid s) > 0 \\ Q_{\min} & \beta(a \mid s) = 0 \end{cases}$$

*Proof.* The objective is

$$\min_Q B(Q) + \alpha \left( \mathbb{E}_{s\sim D, a\sim\mu}(Q(s,a) - Q_{\min})^2 - \mathbb{E}_{s\sim D, a\sim\beta}\frac{\mu(a \mid s)}{\beta(a \mid s)}(Q(s,a) - Q_{\min})^2 \right)$$

which is convex in $Q$.

For a fixed $(s, a)$, consider the first-order optimality conditions in the following cases.

Case 1: $\beta(a \mid s) > 0$.

Note that the derivative of the regularization term vanishes since $\beta(a \mid s) > 0$ and we have

$$\mu(a \mid s)(Q(s,a) - Q_{\min}) - \beta(a \mid s)\frac{\mu(a \mid s)}{\beta(a \mid s)}(Q(s,a) - Q_{\min}) = 0.$$

Thus the optimality condition is the same as the unregularized Bellman objective $\min_Q L(Q) = B(Q)$, the solution of which is $Q^*(s,a) = T^\pi Q(s,a)$.

Case 2: $\beta(a \mid s) = 0$. In this case, the second expectation term has no contribution at $(s, a)$, since $a$ is never sampled under $\beta$. The objective becomes

$$\min_Q B(Q) + \alpha \mathbb{E}_{a\sim\mu}(Q(s,a) - Q_{\min})^2.$$

Taking the derivative with respect to $Q(s,a)$ and setting it to zero yields

$$\frac{dB(Q)}{dQ(s,a)} + 2\alpha\mu(a \mid s)(Q(s,a) - Q_{\min}) = 0.$$

Note that for $\beta(a \mid s) = 0$, $B(Q)$ does not depend on $Q(s,a)$ because the expectation is over in-distribution actions where $\beta(a \mid s) > 0$, so $\frac{dB(Q)}{dQ(s,a)} = 0$ and

$$2\alpha\mu(a \mid s)(Q(s,a) - Q_{\min}) = 0,$$

which implies $Q^\star(s,a) = Q_{\min}$. Combining the two cases, we have

$$Q_{\text{SVR}} = \begin{cases} T^\pi Q(s,a) & \beta(a \mid s) > 0 \\ Q_{\min} & \beta(a \mid s) = 0 \end{cases}$$

$\square$

### A.3   Additional results

In this section, we provide the full results of the evaluation of the variance network behavior model (as described in Section 5.2) and also the Gaussian mixture behavior model (as described in Section 5.3). Table 7 displays these combined results.

Table 7: Full results of learnable $\sigma(s)$ behavior model (DS-var) and Gaussian mixture model (DS-mix), compared to the baseline Gaussian model which uses a fixed $\sigma = 0.2$. SVR results for learnable $\sigma(s)$ behavior model (SVR-var) not shown, due to failure to train.

| Dataset | DS | DS-var | DS-mix |
|---|---|---|---|
| halfcheetah-medium-expert-v2 | 93.7 | 96.0 | 94.5 |
| halfcheetah-expert-v2 | 94.0 | 84.2 | 93.5 |
| hopper-medium-expert-v2 | 110.1 | 110.2 | 103.9 |
| hopper-expert-v2 | 111.1 | 112.1 | 109.9 |
| walker2d-medium-expert-v2 | 109.4 | 110.4 | 90.4 |
| walker2d-expert-v2 | 111.9 | 111.4 | 107.3 |
| halfcheetah-medium-v2 | 63.1 | 59.9 | 61.4 |
| halfcheetah-medium-replay-v2 | 54.6 | 53.7 | 54.3 |
| hopper-medium-v2 | 103.7 | 97.0 | 100.2 |
| hopper-medium-replay-v2 | 103.5 | 99.9 | 104.1 |
| walker2d-medium-v2 | 90.0 | 82.6 | 90.6 |
| walker2d-medium-replay-v2 | 96.2 | 96.5 | 90.4 |
| halfcheetah-random-v2 | 26.8 | 28.3 | 27.5 |
| hopper-random-v2 | 31.0 | 31.1 | 31.4 |
| walker2d-random-v2 | 2.4 | 3.2 | -0.2 |
| gym-v2 total | 1201.6 | 1176.6 | 1159.3 |

