# OpenReview forum: "Density-Scaled Regularization for Offline Reinforcement Learning"
_TMLR — Under review for TMLR_

### Review · Reviewer_ocRk · 2026-05-07

**Summary Of Contributions:**

This paper proposes density scaled (DS) regularization, a new approach for offline reinforcement learning using the conservative principle. The key novelty of DS is the ability to use a behavior policy estimator to scale pessimism applied to the Q function target based on the estimator density. Experiments on state-based and visual D4RL show competitive performance with SOTA methods. Ablations show greater stability than the main benchmark method SVR and reasonable hyperparameter sensitivity.

**Audience:**

Yes

**Audience Explanation:**

Offline RL is a topic of interest to the TMLR audience.

**Claims And Evidence:**

Yes

**Claims Explanation:**

This paper claims that the proposed method penalizes the value function smoothly among other theoretical properties. These properties are theoretically demonstrated. The empirical effectiveness of the proposed method is also demonstrated. My only concern is the usefulness of the method in practice.

**Requested Changes:**

* Tables 1 & 2 demonstrate that DS achieves SOTA performance. However, it also means that DS basically does no better than SVR on these benchmarks. Is there a setting beyond poorly fit behavior policy estimator where DS outperforms SVR or other methods? I understand the contribution of this paper can simply be "whenever people want to use SVR, they should use DS instead" but curious whether there's more to it.
* I am not fully understanding the setup of the robustness experiment in section 5.2. Do you mean you vary the variance of the behavior policy used to generate the data or you vary the variance of the behavior policy estimator that you fit to the data? If the latter then I think the setting is too artificial to be practical: since the purpose of the behavior policy estimator is to provide a good fit to the dataset, I expect the behavior policy class to be maximally expressive (e.g., eq (13)).
* Following the previous comment: table 3 shows that even with this maximally expressive behavior policy class, SVR still failed. Then what kind of behavior policy is used to generate the results in Table 1 and 2?

---

> ### Author Response · Authors · 2026-05-19
> **Response to Reviewer ocRk**
>
> We thank the reviewer for their feedback. Below are our answers to the 3 questions.
> - Section 5.2 shows that DS is preferable to SVR in the setting where the behavior policy is fit with a learned variance, because SVR fails to train with this more expressive policy class - this shows that SVR may require manually constraining parameters to have a stable regularization. In comparison, the performance of our method doesn’t degrade, and we don’t need to tune extra hyperparameters, or introduce tricks to improve numerical stability. We have revised Section 5.2 to emphasize this.
>
> - We tested the latter case (vary the variance of the behavior estimator) in Section 5.2. Indeed, such an experiment is quite artificial, but we treat it as a sensitivity analysis to illustrate how SVR does not adapt well to policies that can return small values of $\beta(a|s)$. The experiment with more expressive behavior policy confirms that such small values can appear in practical and realistic settings. We have revised Section 5.2 to clarify this.
>
> - In Table 1 and 2, the behavior policy with fixed variance is used, i.e. $\pi(a|s) = \mathcal{N}(\mu_{\psi}(s); \sigma=0.2)$. This is the same hyperparameter set in the original SVR method. We have calculated the negative log-likelihood (NLL) of the behavior policy over the dataset against the varying values of $\sigma$:
> | Environment                  | σ=0.02 | σ=0.05 | σ=0.2 | σ=0.8 | σ=2.0 | σ(s)  |
> |------------------------------|-------:|-------:|------:|------:|------:|------:|
> | halfcheetah-expert-v2        |  179.9 |   19.2 |  -2.2 |   4.3 |   9.7 |  -3.3 |
> | halfcheetah-medium-expert-v2 |  209.9 |   24.0 |  -1.9 |   4.3 |   9.7 |  -2.5 |
> | halfcheetah-medium-v2        |  188.1 |   20.5 |  -2.1 |   4.3 |   9.7 |  -3.4 |
> From the above results, the default fixed value of 0.2 achieved competitive NLLs with the learned variance $\sigma(s)$, which can explain the similar performance. We have added a paragraph to discuss this in Section 5.2.

---

> > ### Comment · Reviewer_ocRk · 2026-06-21
> >
> > Thank the authors for their response. I have no further questions.

---

### Review · Reviewer_AUCz · 2026-05-22

**Summary Of Contributions:**

In this paper, the authors tackle the common issue of out-of-distribution (OOD) action overestimation in offline RL by proposing Density-Scaled (DS) regularization. Essentially, they add a penalty to the Bellman update that smoothly scales based on the estimated action density of the behavior policy. What's really neat is the theoretical bridge they build - they show that the existing Supported Value Regularization (SVR) method is basically a limiting case of their DS objective. They also prove that this new DS operator is a valid contraction mapping and its fixed point gives us conservative value estimates that act as a lower bound for the true Q-function.

Strengths:
- Solid Theory: I really appreciated the rigorous math here. The authors do a great job proving how their method sits comfortably between overly pessimistic approaches like CQL and strict support-based methods like SVR.
- Stability and Robustness: The empirical deep dive into the robustness of the behavior density estimator is a highlight. They provide convincing evidence that SVR falls apart and diverges when the behavior model's standard deviation is low, whereas the DS penalty stays rock-solid due to its bounded mathematical formulation.
- Comprehensive Testing: Validating the method on both the standard vector-based D4RL continuous control tasks and the high-dimensional V-D4RL visual tasks really shows off the algorithm's versatility.

 Weaknesses:
- Density Estimation Dependency: Just like SVR, this method heavily leans on having an explicit, learned behavior density model like a Gaussian model. If the offline dataset is super complex or multi-modal, the density estimator might fail to capture the real distribution, which could mess up the value regularization. The authors are aware of this drawback and do acknowledge this as an area for future work.

**Audience:**

Yes

**Audience Explanation:**

Yes, this will be of high interest to the TMLR community. Offline RL is a huge topic right now, especially for trying to get RL working in real-world situations (like healthcare or robotics) where you can't just let an agent explore safely.  The paper strikes a great balance. Theoretically, the audience will appreciate the unifying perspective connecting CQL, SVR, and DS while also appreciating the improved numerical stability and the fact that it works well on complex visual observations in application.

**Broader Impact Concerns:**

There are no major ethical concerns directly stemming from this work that necessitate a detailed Broader Impact Statement. The research focuses on fundamental representation learning, inductive biases, and algorithmic efficiency for spatial planning in simulated RL environments and does not introduce novel harms beyond those standard to the field of robotic navigation and path planning.

**Claims And Evidence:**

Yes

**Claims Explanation:**

- Theoretical Claims: For their claims about bounding overestimation and interpolating existing techniques, the authors provide detailed proofs in the appendix. Specifically, Theorem 1, Proposition 1, and Proposition 2 back up these assertions perfectly.
- Performance: The aggregated benchmark scores in Table 1 (D4RL) and Table 2 (V-D4RL) show DS holding its own—and sometimes slightly beating—SVR and CQL.
- Robustness: Their claim that the bounded penalty ratio is more stable than SVR's unbounded importance ratio is clearly backed up. Figure 1 and Table 3 show SVR suffering from numerical overflow with narrow Gaussian densities (like 0.02 and 0.05) or learnable variance models, while DS trains smoothly under the same conditions.

**Requested Changes:**

The paper acknowledges that it assumes the availability of structured geometry and does not address the construction of this geometry from noisy sensory input. A critical addition would be to include an experiment or an expanded discussion analyzing the robustness of the Hex-PMA architecture to noisy, incomplete, or corrupted triangulations. This would closely mimic the output of real-world SLAM or depth-estimation pipelines and solidify the practical applicability of the approach.

---

> ### Author Response · Authors · 2026-05-27
> **Response  to Reviewer AUCz**
>
> We thank the reviewer for their positive feedback. We would like to point out that the wrong text may have been entered for the requested changes, which talks about something different to our work.

---

### Review · Reviewer_n7Bm · 2026-05-30

**Summary Of Contributions:**

This paper proposes a value-regularization method for offline reinforcement learning to reduce overestimation of out-of-distribution actions, named Density-Scaled (DS) regularization. DS regularization scales the Q-value penalty according to the estimated behavior-policy action density, rather than treating all actions equally or using a hard in-support and out-of-support separation.

This work proves that the DS-regularized Bellman update produces Q-value estimates that lower-bound the true Q-function. It shows that DS regularization can be viewed as a smooth generalization of supported value regularization.

This work conducted extensive experiments on D4RL and visual offline RL benchmarks. The results show that DS regularization is competitive with existing methods such as SVR. The experiments also demonstrate that DS regularization is more numerically stable and less sensitive to behavior-density estimation errors than SVR.

**Additional Comments:**

N/A

**Audience:**

Yes

**Audience Explanation:**

- The paper is likely to be of interest to researchers working on offline reinforcement learning and the robustness of offline actor-critic methods.
- The problem of overestimation of out-of-distribution actions in offline RL is a well-recognized challenge, and the proposed regularization offers a relatively simple modification to existing value-based offline RL algorithms.
- The work bridges two common approaches, including CQL and support-based methods like SVR.

**Broader Impact Concerns:**

I do not see any major broader impact concerns specific to this work beyond those generally associated with offline reinforcement learning.

**Claims And Evidence:**

Yes

**Claims Explanation:**

**Strengths**

- This paper gives a comparison between CQL, SVR, and the proposed DS regularization.
- The theoretical analysis justifies why the proposed penalty interpolates between unregularized in-distribution evaluation and conservative out-of-support value assignment.
- The method is evaluated on standard D4RL benchmarks and visual offline RL benchmarks. The results on D4RL show that DS regularization is competitive with strong offline RL baselines and slightly improves over SVR. Further, experiments support the claim that DS regularization is less sensitive to behavior-density misestimation than SVR.

**Weaknesses**

- The DS regularization underperforms SVR on some Adroit datasets and only slightly improves on the Gym-MuJoCo performance. It would be better to provide an analysis of the differences among these scenarios.
- The robustness study focuses mainly on Gaussian behavior models and selected MuJoCo-style tasks. It would be better to test more density-estimation choices and more environments where the behavior model is misspecified.
- The experiments need to include results like confidence intervals or significance tests, especially because many performance differences are small.

**Requested Changes:**

- The experiments can be strengthened by adding a broader robustness evaluation of the behavior-density estimator. It would be useful to test different density-model classes, such as Gaussian mixtures or normalizing flows.

- A more systematic ablation of the density-scaled penalty would be helpful. For example, the authors could compare the proposed scaling term against alternative bounded density-based penalties, different choices of (Q_{\min}), and variants that use the behavior policy versus the current policy as the penalty sampling distribution.

- It would be better to discuss the applications to broader benchmarks, such as datasets with stronger multimodality and sparse rewards.

---

> ### Author Response · Authors · 2026-06-10
> **Response to Reviewer n7Bm**
>
> We thank the reviewer for their positive feedback. Our responses to the comments are provided below.
>
> ***Weaknesses***
>
> ***W1**. The DS regularization underperforms SVR on some Adroit datasets and only slightly improves on the Gym-MuJoCo performance. It would be better to provide an analysis of the differences among these scenarios.*
>
> We have added a comment in Section 5.1 \- D4RL that addresses this. Note that we have rerun the Adroit experiments by simplifying our DS regularization hyperparameter \- previously we used two sets of parameters (0.04 and 1.0), now we used a fixed parameter of 0.04 and include actor penalization with strength 1.0 across all tasks, as described in the updated main text and experimental details (Appendix). Comment reproduced below:
>
> Note that SVR uses additional hyperparameters specific for the pen-human and pen-cloned environments, where their regularization strength is set to 10, two orders of magnitude higher than their parameter for pen-expert (0.02). In contrast, our DS regularization strength is fixed for all pen environments, which, along with a standard actor penalty term, achieves satisfactory performance without needing to greatly vary the regularization strength for specific environments.
>
> ***W2**. The robustness study focuses mainly on Gaussian behavior models and selected MuJoCo-style tasks. It would be better to test more density-estimation choices and more environments where the behavior model is misspecified.*
>
> See our response to the Requested Changes.
>
> ***W3**. The experiments need to include results like confidence intervals or significance tests, especially because many performance differences are small.*
>
> We have added standard deviations for the benchmark experiments in Table 1 (D4RL) and Table 2 (Vision D4RL). Note that we have had to rerun the benchmarks to generate this, but the results are very similar and our main findings are unchanged.
>
> ***Requested Changes:***
>
> ***R1.** The experiments can be strengthened by adding a broader robustness evaluation of the behavior-density estimator. It would be useful to test different density-model classes, such as Gaussian mixtures or normalizing flows.*
>
> We have included an additional experiment (Section 5.3 \- Mixture density estimator) where we trained a Gaussian mixture model and evaluated our method using it. Our results show that the results are comparable, but the mixture model doesn’t result in performance benefits. We discuss this in the section.
>
> ***R2.** A more systematic ablation of the density-scaled penalty would be helpful. For example, the authors could compare the proposed scaling term against alternative bounded density-based penalties, different choices of (Q\_{\\min}), and variants that use the behavior policy versus the current policy as the penalty sampling distribution.*
>
> We have added additional ablation results (Section 5.3 \- Sampling distribution) that compare variations of the penalty sampling distribution, ranging from the current policy without noise to close to uniform sampling. Our results show that minimal to medium amounts of added noise to the current policy can be beneficial. We don’t consider using the behavior policy as the penalty sampling distribution as the regularizer will not penalize OOD actions in this case.
>
> We’ve only considered the setting of $Q\_{\\min} \= r\_{\\min}/(1-\gamma)$ to ensure that our objective results in a lower bound on the Q-function across all state-action pairs.
>
> ***R3.** It would be better to discuss the applications to broader benchmarks, such as datasets with stronger multimodality and sparse rewards.*
>
> We have added a comment in the Conclusions addressing this:
>
> Our experiments testing a flexible mixture density model provides preliminary evidence that our regularization term can be effective with multi-modal behavior densities. For datasets with stronger multimodality that require more flexible density models, computing the DS regularization term, which involves normalizing by the global maximum of the behavior density, is more challenging and generally intractable. An interesting direction for future work is to consider approximations to the penalty term that can be computed efficiently for general behavior policy models.